# New Work Poses New Challenges—The Importance of Work Design Competencies Revealed in Cluster Analysis

**DOI:** 10.3390/ijerph192114107

**Published:** 2022-10-28

**Authors:** Fiona Niebuhr, Greta M. Steckhan, Susanne Voelter-Mahlknecht

**Affiliations:** Institute of Occupational Medicine, Charité—Universitätsmedizin Berlin, Corporate Member of Freie Universität Berlin and Humboldt Universität zu Berlin, Augustenburger Platz 1, 13353 Berlin, Germany

**Keywords:** New Work, occupational health, work design competencies, working from home, work ability, COVID-19

## Abstract

The continuous transformation process in the world of work, intensified by the COVID-19 pandemic, is giving employees more scope to shape their own work. This scope can be experienced as a burden or as a resource for employees. Work design competencies (WDC) describe employees’ experience of their scope for design. Our study draws on existing datasets based on two Germany-wide studies. We used hierarchical cluster analyses to examine patterns between WDC, the age of employees (range: 18–71 years), the amount of weekly work time working from home (WFH), and work ability. In total, the data of *N* = 1232 employees were analyzed, and 735 of them participated in Study 1. To test the validity of the clusters, we analyzed data from *N* = 497 employees in Study 2. In addition, a split-half validation was performed with the data from Study 1. In both studies, three clusters emerged that differed in age and work ability. The cluster with the highest mean of WDC comprised employees that were on average older and reported a higher mean of work ability. Regarding WFH, no clear patterns emerged. The results and further theoretical and practical implications are discussed. Overall, WDC appear to be relevant to work ability and, in a broader sense, to occupational health, and are related to sociodemographic factors such as age.

## 1. Introduction

The world is in constant transformation, and thus, work as a fundamental part of adult life is also changing. Work has been and continues to be comprehensively transformed by innovation and global competition [1]. Competition accompanied by technological progress is associated with higher levels of uncertainty and velocity of work [2]. Since 2020, the world has been in a state of emergency, experiencing a widespread impact from the COVID-19 pandemic that also affects the working environment. In this context, the pandemic acts as a catalyst for new forms of work and accelerates transformational processes that were already in place before. For instance, telework was rarely used before the pandemic, but rising infection rates required an abrupt switch to *working from home (WFH)* for the majority of employees.

The term New Work is widely used, and yet the definition is vague. The implementation of WFH options and the use of mobile technologies are the most common measures in the context of New Work initiatives in companies [3]. Yet, New Work is not synonymous with working from home; New Work should and must be much more than that. The focus of our study is primarily on the New Work aspect of working from home.

Today, employees are experiencing extensive changes, new demands, more flexibility, and autonomy, and are increasingly required to actively shape their own work. Employees have the opportunity and, at the same time, the obligation to take a more active role in shaping and organizing their work [4]. In this context, newer work design theories often incorporate bottom-up approaches. While managers used to design rather static jobs for their employees via *“top-down” approaches* [5], employees nowadays have a growing opportunity to design their jobs from the *“bottom up”*. In this context, companies should provide the framework conditions for their employees and not shift the whole responsibility for shaping work onto the employees. A common bottom-up approach is the concept of *job crafting*, which was postulated by Wrzesniewski and Dutton [6]. They define job crafting as “the physical and cognitive changes individuals make in the task or relational boundaries of their job” (p. 179). When employees craft their jobs, they behave proactively and voluntarily. Bredehöft et al. [7] studied highly autonomous workers and conceptualized individual work designs less as proactive and voluntary behavior (as in the sense of job crafting), but rather as necessary and reactive behavior. Employees often struggle to design their own work when facing (or given the opportunity of) increased autonomy. This is often the case when WFH, where structures are less hierarchical, and work tasks as well as working hours can be arranged freely.

New forms of work give a lot of possible scope for design. However, the question remains of how competent employees are in using this wider scope for themselves [8]. Against this background, work design competencies gain increasing importance in our new working world. *Work design competencies (WDC)* can be defined as “the knowledge of how to design working conditions in a favorable way that enables effective accomplishment of one’s work tasks while enhancing motivation and reducing stress” [8]. According to Dettmers and Clauß [8], WDC consist of three facets: *planning competency*, *self-motivation competency*, and *stress avoidance competency*. In other words, employees who show higher levels of WDC are presumably able to use increasing scope for design as a resource. They experience less psychological strain and higher motivation. On the other hand, this wider scope for design could be experienced as a burden and excessive demand [8]. While WDC describes one’s own competencies, autonomy is mostly understood as a resource, a framework condition of one’s job. However, especially in professional activities with a high degree of autonomy, it can become an unavoidable demand to make decisions and to structure one’s own work [9,10].

### 1.1. Aim

Due to the COVID-19 pandemic and its catalytic effect on new forms of work, the question remains of who can use the potential of WDC and who perceives the scope for design as a major challenge. To the best of the authors’ knowledge, there are still relatively few studies on WDC. Thus, the aim of the present study is to identify possible clusters between WDC, age, occupational health (i.e., work ability), and working from home (WFH). Another aim of this article is to validate the clusters found in Study 1 in a second study (Study 2) based on a different sample.

### 1.2. Derivation of Test Variables

*Work design competencies* might be based on skills, strategies, and experiences [8]. Employees’ skills and experience presumably increase with years of employment. Older employees might be more experienced in designing their work due to longer employment. However, the current state of knowledge is inadequate. Accordingly, we use *age* of participating employees as a test variable to form clusters. This leads to the following research question:


*RQ1: What patterns emerge regarding WDC and employees’ age?*


Interventions to control COVID-19 infections resulted in a significant increase in the percentage of employees WFH. New scope for design for employees is developed through the rapid change from office to home, the frequent lack of contractual agreements [11], and changes in work groups to virtual teams. Many employees can arrange their working hours freely while WFH, for instance, taking breaks flexibly and determining the sequence of their work tasks. Moreover, the place of work, e.g., hybrid models between WFH and office work, can also be chosen by the employee. We assume that WFH, as one of the biggest changes due to the COVID-19 pandemic towards new forms of work, is providing new resources (e.g., more autonomy), but also imposes high demands on employees. Employees with higher WDC might be able to design their WFH in a more motivating and health-promoting way and might be able to use increased autonomy as a resource. To examine this assumption, we formulate the following second research question:


*RQ2: What patterns emerge regarding WDC and the weekly working hours WFH?*


As a health parameter, we examine the employees’ *work ability* as a test variable to form clusters. Work ability indicates how well individuals are able to work currently and in the near future, and whether employees work in accordance with health resources and work demands [12]. There are numerous studies on potential health effects from WFH. On the one hand, there are health-promoting effects: employees experience fewer interruptions, more privacy, and improved concentration during WFH [13]. On the other hand, WFH working conditions differ compared to working conditions in the office during the pandemic. For example, lower levels of technical equipment due to the rapid change from office to home can have negative effects on work ability and can induce stress-related symptoms [14]. Furthermore, WFH workplaces are often less ergonomically equipped, which can lead to pain or musculoskeletal disorders [15], especially as sedentary time, also detrimental to health, increases during WFH [16]. When the need to plan and decide more independently is perceived as a burden, it might have detrimental effects on wellbeing. High WDC could be understood as a skill to support employees in reducing possible excessive demands [10]. This leads to the following research question:


*RQ3: What patterns emerge regarding WDC and employees’ ability to work?*


## 2. Methods

This study presents post-hoc analyses of existing datasets based on two Germany-wide studies. In the main study (Study 1), participants were interviewed twice as part of the longitudinal study. The presented analysis comprises the subsample of the first measurement point (t1) where data collection took place between July and August 2020. Data from the second measurement point were used in a publication focusing on health effects due to WFH [14]. The second study (Study 2) aims to validate the results of Study 1. The validation of the clusters is intended to replicate the results and to increase generalizability. In the following, we describe the explorative study design and the procedure, as well as the used measurement instruments. Finally, we provide information about the statistical analysis and the calculated cluster analyses. In their review, Clatworthy et al. [17] examined 59 published cluster analyses from the research field of health psychology. We followed their derived guidelines regarding the relevant information that should be reported when conducting a cluster analysis.

### 2.1. Sample and Procedure

#### 2.1.1. Longitudinal Sample (Study 1)

The online survey of the first study was conducted by a panel company. Participants lived in Germany, were employed, and worked part of their weekly working hours from home. They received a small expense allowance for participating in the study. The survey was conducted in German and items were translated for this article by the research team. In total, *N* = 735 employees participated fully at Time 1. The average age was 44.32 (*SD* = 12.74), ranging from 18 to 69 years. The sample was relatively balanced in terms of gender, 51% were female and nobody reported being diverse.

#### 2.1.2. Cross-Sectional Sample (Study 2)

Employees in a large German hospital (*n*_1_ = 585) and in a large company for industry, infrastructure, and mobility (*n*_2_ = 100) were surveyed via an online survey as part of a Germany-wide COVID-19 study by the research team. Overall, *N* = 497 participants (*n*_1_ = 429; *n*_2_ = 68) completed the survey. We excluded participants from the analyses who did not complete the whole survey. The mean age of participants was 42.96 years (*SD* = 11.18), ranging from 19 to 71 years. The gender ratio was rather unbalanced, with 69% female participants; 4 (0.5%) participants defined themselves as diverse.

### 2.2. Measures

The *percentage of weekly working hours WFH* was measured using a single item developed by the research team. Participants were asked to state on a sliding scale ranging from 0–100% what percentage of their weekly work time they currently worked from home.

Participants rated their *work ability* on a 10-point scale, ranging from *completely unable to work* (0) to *best work ability* (10). This item originates from the *Work Ability Index (WAI)* by Hasselhorn and Freude [18]. The use of this single item as the *Work Ability Score (WAS)* is valid and is commonly used in occupational health research [12].

Dettmers and Clauß [8] developed a scale to assess *work design competencies*. We applied the complete scale consisting of 11 items; 5 items were used to measure *planning competence* (*PC*; e.g., “When you think about your work, how well do you manage to structure your tasks by yourself?”) and 3 items to capture *self-motivation competence* (*SMC*) as well as *stress avoidance competence* (*SAC*), respectively [8]. In our sample, internal consistencies were α = 0.91; α_PC_ = 0.89, α_SMC_ = 0.81, and α_SAC_ = 0.79. This reliability estimate is comparable to the reported values in the validation study [8]. In addition, participants were asked to report their *age* and *gender*.

### 2.3. Data Analysis

Following the guidelines of Clatworthy et al. [17], we report five types of information below.

#### 2.3.1. Computer Program

All analyses were performed using IBM SPSS Statistics 26.0 (IBM, Armonk, NY, USA).

#### 2.3.2. Similarity Measure

The choice of similarity measure can have significant effects on cluster analysis. For our study, it is important to consider the scores of individuals on the test variables, as well as differences in levels. Therefore, we use *Squared Euclidean distance* to group like-minded individuals. In Euclidean distance, two cases are compared per variable. The algorithm allows the distance between two cases to be computed across all variables and to be reflected in a single distance value [19]. This similarity measure is commonly used in health psychology [17].

#### 2.3.3. Cluster Method

The main difference between the two basic methods of cluster analysis, hierarchical and nonhierarchical, is that hierarchical cluster analyses do not form an initial set of cluster means [19]. Therefore, due to the exploratory design of our study, we conducted a hierarchical cluster analysis. In this, either a new cluster is formed in each step, or a case is assigned to the previous cluster [19]. We conducted a hierarchical agglomerative approach and used the *Ward method* [20]. In this method, a cluster is first formed for each case. The cases are then regrouped until the variance within each cluster reaches a defined minimum level [21].

#### 2.3.4. Procedure Used to Determine the Number of Clusters

The hierarchical cluster analysis approach produces in a first step as many clusters as cases [17]. We explored the appropriate number of clusters using an agglomeration schedule as well as a dendrogram. Proportionally large increases from one coefficient to another indicate that the clusters are distinct at this point. Consequently, the clustering process should be terminated at the previous stage. Based on these criteria, we selected three clusters in each cluster analysis.

#### 2.3.5. Evidence for the Validity of the Clusters

Simply speaking, even homogenous cases can be grouped into different clusters using the algorithm. To ensure that the cluster analysis shows the structure in the data and does not create it itself, the results of the cluster analysis should be validated [17]. To warrant that the cluster analysis we report here is valid, we performed two different validations. A common method is to split the sample into randomized halves and run the cluster analysis again in each half. Valid clusters should show similar patterns in both sample halves [17]. Finally, we conducted the cluster analysis with a different data set, the sample of the cross-sectional study of German employees.

## 3. Results

### 3.1. Study 1

For comparability of results from the studies and cluster analyses reported here, we used consistent cluster numbering. The numbering is in descending order based on the values of WDC. Table 1 presents means, standard deviations, and bivariate correlations of all variables included in the first cluster analysis.

As a first step, we performed a hierarchical cluster analysis applying Ward’s method using test variables, age, amount of weekly working time WFH, ability to work, and WDC. Three clusters were selected using the dendrogram and the agglomeration schedule. Second, we performed parametric group comparisons to describe the three clusters and examined significant differences among them, as can be seen in Table 2. In addition, Table 2 shows the descriptive characteristics of participants within the three clusters based on the test variables used.

The assumptions for the *analysis of variance* (*ANOVA*) were checked prior to the tests. In all of them, the variance homogeneity requirement was not fulfilled. Accordingly, the Welch ANOVA is reported, and the Games–Howell post-hoc test was used. As depicted in Table 2, participants in the three clusters differed significantly in their reported work ability. Mean level of work ability decreased significantly from Cluster 1 to Cluster 2 and from Cluster 2 to Cluster 3. Furthermore, there was a significant decrease of mean level of WDC from Cluster 1 to Cluster 3 and from Cluster 2 to Cluster 3. Thus, participants in Cluster 1 reported, on average, the highest WDC as well as the highest mean level of work ability, whereas participants in Cluster 3 showed, on average, the lowest WDC and the lowest mean level of work ability. In addition, WDC and work ability showed a moderate significant positive association in Table 1. Furthermore, we found significant differences between all clusters in terms of age, as well as in the amount of work time per week WFH, as can be seen in Table 2. The results of the Games–Howell post-hoc analysis revealed that participants in Cluster 3, who reported, on average, the lowest mean of WDC, are significantly younger than individuals in Cluster 1 and 2. In addition, employees within Cluster 1, comprising participants with the highest mean level of WDC, work significantly more from home than employees in Cluster 2 and Cluster 3.

### 3.2. Evidence for the Validity of the Clusters

We randomly divided the sample into two halves using the corresponding function in SPSS to identify the stability of clusters. In the following, we present the results of the data from the first half, followed by those from the second half.

#### 3.2.1. Split-Half

In total, *n* = 366 participants were in the first half of the total dataset and were clustered when the hierarchical cluster analysis was conducted using Ward’s method. Once again, three clusters were found based on the evaluation of the dendrogram and the agglomeration schedule.

In line with the results mentioned above, employees within the three clusters differed significantly with respect to WDC, work ability, and age. Table 3 presents the results of the parametric group comparisons in detail, which differed slightly compared to the first cluster analysis. The average amount of weekly working time WFH differed significantly between all clusters according to the Games–Howell post-hoc test. It decreased from Cluster 1 to Cluster 3 and from Cluster 3 to Cluster 2.

There were 369 participants in the second half of the dataset that were assigned to clusters using Ward’s method. The evaluation of the agglomeration schedule and the dendrogram also resulted in a three-cluster solution. Welch ANOVA values and, accordingly, the Games–Howell post-hoc analysis are reported for all variables due to missing variance homogeneity. The patterns of the first cluster analysis regarding WDC, work ability, and age could be replicated. The detailed results are presented in Table 4. The average amount of work WFH was significantly higher in Cluster 2 compared to WFH in Cluster 1 and Cluster 3.

The screening of participant IDs from Cluster 1 in each (sub)dataset revealed high levels of similarity, i.e., the majority of participants classified in Cluster 1 (complete sample) are also grouped in Cluster 1 (first half of the data) or Cluster 1 of the second half of the data.

#### 3.2.2. Study 2

To further test the stability of the three clusters found here, we conducted the analysis again with another sample from a cross-sectional study. Table 5 presents means, standard deviations, and bivariate correlations of all variables included.

Cluster analysis was performed analogously to the analysis reported above in terms of replication. The same test variables were used for this purpose, which were surveyed by using identical items. The full sample of *N* = 497 cases were grouped into clusters. Based on the visual inspection of the dendrogram and the increase in coefficients, three clusters emerged. In each case, the coefficients in the assignment summary (SPSS output) indicate the distance at which a merge of the two clusters to the left occurs. The descriptive characteristics regarding the test variables within the three clusters, as well as the result of analyses of variance and post-hoc tests, are presented in Table 6.

In all parametric group comparisons performed, the requirement of homoscedasticity was not fulfilled. Accordingly, we report the Welch ANOVA values and performed analyses using the Games–Howell post-hoc test. We found significant differences between the three clusters for all test variables. The mean level of WDC decreases significantly from Cluster 1 to Cluster 2 and from Cluster 2 to Cluster 3, as depicted in Table 6. Mean work ability is significantly lower in Cluster 3 than in Cluster 1 and in Cluster 2. Regarding age, the Games–Howell post-hoc analysis showed that employees in Cluster 3 are significantly younger than those in Cluster 1 as well as individuals in Cluster 2. Employees in Cluster 2 worked significantly less time from home than individuals in Cluster 1 and in Cluster 3.

Across the different samples, our results revealed a similar pattern regarding WDC, work ability, and age, whereas a consistent pattern between WDC and WFH was not found. A higher mean value of WDC came along with a higher mean level of work ability and a higher average age. Furthermore, WDC showed a significant positive association with work ability and age in both samples, as shown in Table 1 and Table 5.

## 4. Discussion

The continuous transformation process in the world of work, accelerated by the COVID-19 pandemic, gives employees more scope to shape their own work. At the same time, this scope for design can be experienced as a burden and thus may not be available as a resource for employees. The competence of employees to shape their own work and therefore take advantage of the new scope for design is a decisive factor for New Work. The aim of this study was to explore the patterns between these work design competencies (WDC), the age of employees, the percentage of weekly working hours WFH, and work ability using cluster analyses.

Regarding our first research question (RQ1), cluster analyses in Studies 1 and 2, as well as the validation with the dataset halves, revealed a consistent pattern: older employees reported higher WDC on average than younger employees. Based on the concept of competence, it can be assumed that older employees have acquired the skills and strategies to shape their own work due to several years of work experience. Accordingly, older employees showed higher levels of WDC compared to younger employees.

Related to the second research question (RQ2), we found in the samples of both studies a positive significant association between the percentage of weekly time WFH and WDC. However, the clusters do not show a clear pattern in this regard. In Study 1, employees in Cluster 1 with the highest mean of work ability and WDC had the highest amount of worktime WFH on average compared to employees in Cluster 2 and Cluster 3. Surprisingly, participants in Cluster 3, characterized by the lowest mean of work ability and WDC, also showed a higher mean level of time WFH than employees in Cluster 2. A similar, hardly interpretable, pattern was found in Study 2. While employees in Cluster 1 reported the highest mean of work ability and WDC and clearly differed in this regard from employees in Cluster 3, both groups reported on average a higher level of weekly working time WFH. Employees in Cluster 2, who also showed a higher mean of work ability and WDC than Cluster 3, worked only 33% from home. It could be argued that employees with a higher percentage of WFH have more experience in using the scope for design. Accordingly, they may be more structured, work more independently, and use the freedom in shaping their work compared to employees who do not practice, or practice less, WFH. Thus, it would be conceivable that this group of employees might also have a higher level of WDC. However, the amount of weekly working time WFH in Cluster 3, comprising employees that showed the lowest mean level of WDC, remains difficult to interpret. Longitudinal studies are required to further examine the interplay of WDC and the weekly working time WFH.

With regard to research question 3 (RQ3) focusing on work ability, we found similar patterns in both Study 1 and Study 2, as well as in the split-half validations. Employees in Cluster 1 reported the highest mean of work ability; in line with WDC work ability decreased from Cluster 1 over Cluster 2 to Cluster 3 in all cluster analyses. In addition, a similar pattern to RQ1 was found: Cluster 3 with the lowest mean of work ability comprised employees with the youngest average age. This surprising pattern was also found in the validation studies. Cluster analyses with the randomly split dataset showed similar patterns in both halves of the data. Cluster 1 and Cluster 2 comprised employees with a higher mean age that reported higher mean levels of work ability compared to employees in Cluster 3. This contradicts previous empirical findings. Van den Berg et al. [22], for example, examined the relationship between individual factors and work ability in a review based on 20 studies. Their results suggest that older age was associated with weaker work ability. It is possible that the pattern found here could be explained by an interaction of age and WDC. WDC might moderate and compensate for the negative relationship between age and work ability; however, this assumption could not be confirmed with our data as age showed a positive association with work ability in Study 2. In a similar moderation model, Weigl et al. [23] found that the three action strategies from the *SOC model* [24]—*selection, optimization, and compensation*—reduce the negative association between age and work ability. The association was weakest for employees who reported high job control and frequently used SOC strategies [23]. In our study, employees grouped in Cluster 1 reported the highest mean of work ability and WDC, as well as a higher mean age, compared to employees in Cluster 3. To examine and expand the results of Weigl et al. [23] the interaction of these factors should be investigated in future studies.

### 4.1. Strengths and Limitations

A strength of the present study is the diversity of the sample in Study 1 in terms of industries, age, and occupations. This sample represents different employee groups, industries, income classes, educational backgrounds, and employment statuses of the German working population. In contrast, the sample in Study 2 is very specific and represents two companies and branches in Germany. Hence, our results apply to a diverse sample with different employee groups as well as to a rather specific sample. Moreover, the groups formed on the basis of the clusters in both studies are sufficiently large, increasing the generalization of the current findings.

A further strength is that the results of the first cluster analysis were validated twice. In addition to validation using randomized data halves, a further sample from a cross-sectional study was used. The survey dates of both studies were close in time. The patterns found in the first cluster analysis were thus validated several times and with different samples, which enhances the generalization of our study.

Furthermore, to the best of our knowledge, there are very few studies on WDC so far, although they are of high importance in the context of New Work. The exploratory design of our work allows us to gain initial insights on the relevance of WDC during the COVID-19 pandemic. In addition, we used prescribed standards in conducting and reporting the cluster analyses. Clatworthy et al. [17] developed these based on reviewing different cluster analyses in the field of health psychology. Using this approach strengthens the present work.

As with any field study, there are some limitations that need to be addressed. Regarding older employees that reported more pronounced WDC, there may be confounding effects. It is conceivable that employees with lower WDC had fallen ill or left working life for other reasons before data collection started. Future longitudinal studies should conduct dropout analyses to consider this potential bias. Furthermore, participating employees in both studies worked part of their weekly work hours from home. It seems plausible that WFH provides many possibilities for shaping one’s own work and at the same time imposes high demands on employees. The extent of individual WDC could play a central role here, not least to protect employees’ health while WFH. Regarding the percentage of working time WFH, no clear pattern emerged in the cluster analyses. In the current study, the factor was surveyed by a single item. Future longitudinal studies should measure WFH in a more complex, multifactorial design and examine associations with WDC.

In the present study, patterns between WDC and the amount of work time WFH were examined. WFH can be considered as a characteristic of work. Future studies could examine the extent to which WDC is associated with different work characteristics (e.g., social support, autonomy) and which manifestations of work characteristics support employees in utilizing and strengthening their individual WDC. In a recent study by Mishima-Santos et al. [25], the authors investigated the patterns between work characteristics of remote work and employee wellbeing. Among others, the variables decision and execution autonomy made the greatest contribution to the clusters [25]. It might be assumed that a certain degree of autonomy at work strengthens WDC.

Accordingly, the samples used in this study were rather homogeneous with respect to the participants’ places of work. Future studies should examine other work locations and should draw comparisons between WFH, office work, or hybrid models.

Due to the scarcity of studies on WDC and health associations in the context of New Work, we chose an exploratory design. Future studies should choose other study designs to gain new insights. Longitudinal analyses, for instance, could examine the direction of associations between WDC, work ability, and WFH. Finally, the data used here originate from a single source and are self-reported; this carries the risk of social desirability bias.

### 4.2. Practical Implications

The clusters found are an indication that, older workers may have more pronounced WDC than younger workers. These competencies might strengthen the work ability of older workers. It seems plausible that with increasing work experience, employees are able to use and shape creative scope for design. Higher age and correspondingly more work experience could therefore strengthen WDC. Interaction effects between WDC, age, and work ability were not the focus of the present study but are conceivable and a possible explanation for the patterns found here.

Teams that consist of employees of different age groups could benefit from the experience of older colleagues that might increase WDC of all employees. Prejudice against diversity can inhibit satisfaction and productivity in diverse teams. Ageism could emerge more frequently, especially in the context of New Work and digital work, with the intensive use of new technologies and increasingly flexible workplaces. Paoletti et al. [26] point out that teams are a key success factor for organizations and that the empirical findings to date regarding age diversity in teams are ambivalent. The authors argue that the salience of age diversity is associated with negative outcomes such as ageism and burnout. Talking openly about positive components of aging can help in this regard [26]. Accordingly, age-diverse teams could benefit from the experience of older workers, and informative workshops about WDC and the impact of experience on these and other competencies could reduce conflicts in diverse teams. Furthermore, our results show that WDC came along with higher work ability. This finding emphasizes the importance of strengthening WDC against the background of occupational health.

In addition, companies should consider the fact that employees might want varying degrees of scope for design. Agile culture and methods as well as flat hierarchies could support employees in making use of the given scope. Interventions to promote individual WDC could strengthen employees who tend to perceive current scope for design as a burden and are not (yet) able to use it as a resource.

## 5. Conclusions

The world of work is in a constant state of transformation, and there is increasing scope for employees to shape their own work. Agility, flexibility, and autonomy—some aspects of New Work—can be resources for employees. At the same time, this scope for shaping their own work can be experienced as a burden and thus might have adverse health effects. In the context of New Work, WDC are relevant, and the question arises of how competent employees are in shaping their own work. This article provides initial indications that WDC are relevant for work ability and in a broader sense for occupational health and are related to sociodemographic factors such as age. In the future, the world of work will continue to change, and new forms of work will become a reality for many of us. Accordingly, the required competencies need to be researched and promoted for healthy New Work.

## Figures and Tables

**Table 1 ijerph-19-14107-t001:** Descriptive statistics and bivariate correlations of test variables (*N* = 735).

		*M*	*SD*	1	2	3	4
1	Age	44.32	12.74	1.00			
2	Amount of worktime WFH	56.74	39.25	.01	1.00		
3	Work ability	8.68	1.96	.08	.09 *	1.00	
4	Work design competencies	3.78	0.69	.23 **	.12 **	.37 **	1.00

Note: ** *p* < .01, * *p* < .05.

**Table 2 ijerph-19-14107-t002:** Descriptive statistics of the test variables divided into clusters (*N* = 735), analyses of variance, and post-hoc tests.

Measure	Cluster 1		Cluster 2		Cluster 3		*F*	*df*	*p*	*η* ^2^	Post-Hoc Test		
	*M*	*SD*	*M*	*SD*	*M*	*SD*					Comparison	Mean Difference	*p*
Age	48.59	10.93	52.99	8.35	37.32	11.80	151.74	2, 435	.001 ***	.27	Cluster 1 vs. 2	4.40	.001 ***
											Cluster 2 vs. 3	15.68	.001 ***
											Cluster 3 vs. 1	11.27	.001 ***
Amount of worktime WFH	90.25	16.01	20.25	20.80	49.07	38.73	677.81	2, 402	.001 ***	.44	Cluster 1 vs. 2	70	.001 ***
											Cluster 2 vs. 3	28.82	.001 ***
											Cluster 3 vs. 1	41.18	.001 ***
Work ability	9.76	1.07	9.43	1.14	7.56	2.16	133.29	2, 428	.001 ***	.28	Cluster 1 vs. 2	0.33	.012 *
											Cluster 2 vs. 3	1.87	.001 ***
											Cluster 3 vs. 1	2.20	.001 ***
Work design competencies	4.17	0.52	4.13	0.44	3.35	0.62	186.06	2, 431	.001 ***	.35	Cluster 1 vs. 2	0.04	**.668**
											Cluster 2 vs. 3	0.78	.001 ***
											Cluster 3 vs. 1	0.82	.001 ***

Note: *N*_Cluster 1_ = 244, *N*_Cluster 2_ = 153, *N*_Cluster 3_ = 338; * *p* < .05, *** *p* < .001; non-significant values are printed in bold.

**Table 3 ijerph-19-14107-t003:** Descriptive statistics of the test variables divided into clusters (*N* = 366), analyses of variance, and post-hoc tests.

Measure	Cluster 1		Cluster 2		Cluster 3		*F*	*df*	*p*	*η* ^2^	Post-Hoc Test		
	*M*	*SD*	*M*	*SD*	*M*	*SD*					Comparison	Mean Difference	*p*
Age	51.80	7.98	50.00	10.78	34.20	9.35	151.14	2, 219	.001 ***	.44	Cluster 1 vs. 2	1.80	**.360**
											Cluster 2 vs. 3	15.80	.001 ***
											Cluster 3 vs. 1	17.61	.001 ***
Amount of worktime WFH	88.30	16.5	15.27	16.73	65.89	34.5	504.16	2, 240	.001 ***	.54	Cluster 1 vs. 2	73.03	.001 ***
											Cluster 2 vs. 3	50.62	.001 ***
											Cluster 3 vs. 1	22.41	.001 ***
Work ability	9.77	1.07	9.16	1.37	7.84	2.18	45.08	2, 240	.001 ***	.19	Cluster 1 vs. 2	0.61	.001 ***
											Cluster 2 vs. 3	1.32	.001 ***
											Cluster 3 vs. 1	1.93	.001 ***
Work design competencies	4.16	0.47	4.01	0.66	3.41	0.57	63.27	2, 363	.001 ***	.26	Cluster 1 vs. 2	0.15	**.143**
											Cluster 2 vs. 3	0.60	.001 ***
											Cluster 3 vs. 1	0.75	.001 ***

Note: *N*_Cluster 1_ = 96, *N*_Cluster 2_ = 108, *N*_Cluster 3_ = 162; *** *p* < .001; non-significant values are printed in bold.

**Table 4 ijerph-19-14107-t004:** Descriptive statistics of the test variables divided into clusters (*N* = 369), analyses of variance, and post-hoc tests.

Measure	Cluster 1		Cluster 2		Cluster 3		*F*	*df*	*p*	*η* ^2^	Post-Hoc Test		
	*M*	*SD*	*M*	*SD*	*M*	*SD*					Comparison	Mean Difference	*p*
Age	45.93	13.41	47.48	11.71	40.17	12.92	10.38	2, 207	.001 ***	.05	Cluster 1 vs. 2	1.56	**.585**
											Cluster 2 vs. 3	7.31	.001 ***
											Cluster 3 vs. 1	5.76	.006 **
Amount of worktime WFH	36.59	31.04	92.26	14.30	17.04	22.95	502.05	2, 173	.001 ***	.68	Cluster 1 vs. 2	55.67	.001 ***
											Cluster 2 vs. 3	75.22	.001 ***
											Cluster 3 vs. 1	19.55	.001 ***
Work ability	9.83	0.92	8.72	1.68	7.07	2.48	65.37	2, 198	.001 ***	.25	Cluster 1 vs. 2	1.11	.001 ***
											Cluster 2 vs. 3	1.64	.001 ***
											Cluster 3 vs. 1	2.75	.001 ***
Work design competencies	4.17	0.46	3.83	0.66	3.21	0.67	69.89	2, 217	.001 ***	.26	Cluster 1 vs. 2	0.34	.001 ***
											Cluster 2 vs. 3	0.62	.001 ***
											Cluster 3 vs. 1	0.96	.001 ***

Note: *N*_Cluster 1_ = 109, *N*_Cluster 2_ = 166, *N*_Cluster 3_ = 94; ** *p* < .01, *** *p* < .001; non-significant values are printed in bold.

**Table 5 ijerph-19-14107-t005:** Descriptive statistics and bivariate correlations of test variables (*N* = 497).

		*M*	*SD*	1	2	3	4
1	Age	42.75	11.66	1.00			
2	Amount of worktime WFH	72.91	30.62	−.11 *	1.00		
3	Work ability	8.11	1.62	.18 **	−.02	1.00	
4	Work design competencies	3.67	0.56	.13 **	.09 *	.47 **	1.00

Note: ** *p* < .01, * *p* < .05.

**Table 6 ijerph-19-14107-t006:** Descriptive statistics of the test variables divided into clusters (*N* = 497), analyses of variance, and post-hoc tests.

Measure	Cluster 1		Cluster 2		Cluster 3		*F*	*df*	*p*	*η^2^*	Post-Hoc Test		
	*M*	*SD*	*M*	*SD*	*M*	*SD*					Comparison	Mean Difference	*p*
Age	44.54	11.06	45.74	10.82	34.20	6.96	64.97	2, 246	.001 ***	.13	Cluster 1 vs. 2	1.2	**.523**
											Cluster 2 vs. 3	11.54	.001 ***
											Cluster 3 vs. 1	10.33	.001 ***
Amount of worktime WFH	90.16	12.87	32.51	17.98	91.34	17.27	628.84	2, 182	.001 ***	.75	Cluster 1 vs. 2	57.65	.001 ***
											Cluster 2 vs. 3	58.83	.001 ***
											Cluster 3 vs. 1	1.18	**.840**
Work ability	8.60	1.23	8.36	1.32	6.33	1.97	47	2, 181	.001 ***	.25	Cluster 1 vs. 2	0.241	**.159**
											Cluster 2 vs. 3	2.03	.001 ***
											Cluster 3 vs. 1	2.27	.001 ***
Work design competencies	3.9	0.46	3.58	0.54	3.07	0.44	107.55	2, 205	.001 ***	.28	Cluster 1 vs. 2	0.32	.001 ***
											Cluster 2 vs. 3	0.51	.001 ***
											Cluster 3 vs. 1	0.83	.001 ***

Note: *N*_Cluster 1_ = 265, *N*_Cluster 2_ = 153, *N*_Cluster 3_ = 79; *** *p* < .001; non-significant values are printed in bold.

## Data Availability

The data presented in this study are available on reasonable request from the corresponding author, F.N. The data are not publicly available, as this was assured to the participants in the study information as well as in the data privacy statement.

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
