# Peer review of "New Work Poses New Challenges—The Importance of Work Design Competencies Revealed in Cluster Analysis"

_ijerph, 2022, doi:10.3390/ijerph192114107_

Round 1
Reviewer 1 Report
Dear author,
I read with interest your work about work design competencies in the light of the New Work, and the post-pandemic era.
I hope my comments will help you.
-aim: I suggest to explain better the aims, it is not so clear, for example you can state the research questions or hypothesis. Also I would appreciate a cleare explanation regarding the use of two different samples, why was it necessary?
-Results: I strongly recommend to make the results paragraph more readable, you can use tables that include the results of the statistical analysis and you can provide a summary at the end of each study. IN other words, at this moment, Results paragraph is very hard to understand and to follow.
-Discussion needs improvements in terms of comparing your findings with pre-existent literature. It would be interesting to point out if in similar studies before pandemic there was something different from your findings.
Best regards
Author Response
Dear Reviewer,
thank you for allowing us to revise our manuscript. The reviewers’ careful comments are much appreciated, and they have helped us to strengthen the manuscript. We have responded to the reviewers’ points below and revised our manuscript accordingly using track changes.
In summary, we have considered the editorial comments, included additional information in the methods section, clearly structured the results section, included tables, and shortened the written presentation of the results. We have addressed all of your comments, which are described in detail below.
Please see the attachment.

Reviewer 2 Report
Dear authors,
Thank you for the opportunity to read and comment on your work.
I would like to congratulate the authors of the theme of your work, and the comments that I will do here will follow the reading process. Thus, I will comment on the lines and I hope I could offer “food for thought”.
Lines 47-58: In general, the authors have a reason, but the critical literature will say that the organizations will “pull out” and do not assume what they have and should do based on the argument that the worker should redesign his/her work. In other words: it helps to blame the victim.
Lines 60-74: Interestingly the WDC with its three facets: “planning competency, self-motivation competency, and stress avoidance competency” are similar to what Mishima-Santos et al. 2021 found in their Work Design in Teleworkers.
Lines 100-116: The authors have two research questions: (1) WDC and age (2) WDC and weekly working time as a proxy for health.
Although the questions make sense, there are few evidence of the gap in the questions. Because if there are no issues about it, in other words, if it is expected with no doubt that WDC increases with age, what is the point of testing it? In the study mentioned there are no differences in age, for example. So, this kind of gap should be included.
The question about work ability was not stated.
However, more than anything, why are those questions answered using cluster analysis?
I suggest before the method section the inclusion of a research scheme to be tested, a figure.
Method and results:
Cluster analysis is a method that for Likert scales works better using log-likelihood. It allows the normalization of the scale. Why don’t the authors describe the silhouette of the cluster? This describes the quality of the cluster as well.
Why Table 1 describes “work design competencies” in general and not each variable separately and in the method the authors should describe the internal consistency (alpha or omega or Guttman or?) for each factor.
Although I agree with the authors that split-half could be important to validate the general idea, all this 3.2 section could be in a complementary file.
Again, what is the issue about Study 2 that was not originally described in the introduction section and the aim of the work?
What is the contribution of each variable to the cluster? SPSS describes it. Part of that calculus of ANOVA could be removed with this evidence.
Discussion:
There is a question not answered: older employees have more WDC or are they survivors? Because if they don’t have WDC they may be sick or unemployed. Thus, what we was measured at the end was survival.
About work ability: as the authors did not describe the strength of the variable in the cluster it is difficult to know how much this variable is that important there. The strength could help describe and give more information.
One last point that is important to think: work design competencies are relevant for people health or for companies? The main argument is that older people may be relevant for organizations? People that work less (healthier?) are those who describe average work design competencies and in general elderly (older than the other groups). I mean, the point of the manuscript is not clear at the end, and it could be clarified by the authors.
Author Response
Dear Reviewer,
thank you for allowing us to revise our manuscript. The reviewers’ careful comments are much appreciated, and they have helped us to strengthen the manuscript. We have responded to your points below and revised our manuscript accordingly using track changes.
In summary, we have considered the editorial comments, included additional information in the methods section, clearly structured the results section, included tables, and shortened the written presentation of the results. We have addressed all of your comments, which are described in detail below.
Please see the attachment.

Reviewer 3 Report
I applaud the authors for the quality of research done in the study. The manuscript very much contributed to the changing world of work.
Please attend to the minor correction (editing) as highlighted in the reviewed manuscript.

Author Response
Dear Reviewer,
thank you for your time and for allowing us to revise our manuscript. The reviewers’ careful comments and editing are much appreciated, and they have helped us to strengthen the manuscript. We have responded to the reviewers’ points below and revised our documents accordingly using track changes.
Please see the attachment.
